# Iterative Nash Policy Optimization: Aligning LLMs with General Preferences via No-Regret Learning

**Yuheng Zhang** [*1,2]  **Dian Yu** [2]  **Baolin Peng** [2]  **Linfeng Song** [2]  **Ye Tian** [2,3]
**Mingyue Huo** [1]  **Nan Jiang** [1]  **Haitao Mi** [2]  **Dong Yu** [2]
[1]University of Illinois Urbana-Champaign    [2]Tencent AI Lab, Bellevue, WA
[3]Tencent Robotics X
{yuhengz2,mhuo5,nanjiang}@illinois.edu
{yudian,baolinpeng,lfsong,haitaomi,dyu}@global.tencent.com
{yaptian}@tencent.com

## Abstract

Reinforcement Learning with Human Feedback (RLHF) has achieved great success in aligning large language models (LLMs) with human preferences. Prevalent RLHF approaches are reward-based, following the Bradley-Terry (BT) model assumption, which may not fully capture the complexity of human preferences. In this paper, we explore RLHF under a general preference framework and approach it from a game-theoretic perspective. Specifically, we formulate the problem as a two-player game and propose a novel online algorithm, iterative Nash policy optimization (INPO). The key idea is to let the policy play against itself via no-regret learning, thereby approximating the Nash policy. Unlike previous methods, INPO bypasses the need for estimating the expected win rate for individual responses, which typically incurs high computational or annotation costs. Instead, we introduce a new loss objective that is directly minimized over a preference dataset. We provide theoretical analysis for our approach and demonstrate its effectiveness through experiments on various representative benchmarks. With an LLaMA-3-8B-based SFT model, INPO achieves a 42.6% length-controlled win rate on AlpacaEval 2.0 and a 37.8% win rate on Arena-Hard, showing substantial improvement over the state-of-the-art online RLHF algorithms.

## 1 Introduction

Large language models (LLMs) such as ChatGPT (Achiam et al., 2023), Claude (Anthropic, 2023), and Bard (Google, 2023) have achieved tremendous success in various instruction-following tasks. A key factor in this success is the technique of reinforcement learning with human feedback (RLHF) (Christiano et al., 2017), which aligns LLMs with human preferences and values. The first standard RLHF framework for LLM alignment was proposed by Ouyang et al. (2022). They first train a reward model (RM) on a dataset containing human preferences. Subsequently, a pretrained LLM is fine-tuned to maximize the reward from this RM using the proximal policy optimization (PPO) algorithm (Schulman et al., 2017). Models trained with this pipeline can generate human-preferred outputs even with 100x fewer parameters. Nevertheless, fitting a high-quality RM requires a large amount of human-labeled data, and training with PPO is generally less stable (Peng et al., 2023). To bypass the training of the RM, Rafailov et al. (2024) propose the direct preference optimization (DPO) algorithm, which directly learns a policy on a human preference dataset. Compared to RLHF with PPO, DPO is more stable and computationally lightweight.

However, the approaches mentioned above, which rely on either an explicit or implicit RM, assume that human preferences can be adequately modeled with the Bradley–Terry (BT) model (Bradley & Terry, 1952). We argue that the BT model cannot fully capture the complexity of human preferences. For example, the preference signal in the BT model is transitive, implying that if $A$ is preferred

---

*Work done during an internship at Tencent AI Lab, Bellevue, WA.

to $B$ and $B$ is preferred to $C$, $A$ must be preferred to $C$. This kind of transitive property may not always hold across diverse human groups and contradicts evidence in human decision-making (May, 1954; Tversky, 1969). In addition, experimental results show that the accuracy of BT-based RMs is about 70% (Bai et al., 2022c; Cui et al., 2023), while preference models outperform them by a clear margin (Ye et al., 2024). This motivates us to consider general preferences without the BT model assumption.

To achieve this goal, Munos et al. (2023) formulate the LLM alignment problem as a symmetric two-player game. One can show that for any other policy, the Nash policy of the game enjoys at least one half win rate, ignoring the KL regularization terms. Given the general preference oracle, Munos et al. (2023) propose a ***planning*** algorithm to solve for the Nash policy. In this paper, we consider the ***learning*** problem, where the general preference oracle is unknown to us, and we only assume access to query the oracle. Inspired by the connections between constant-sum games and online learning (Freund & Schapire, 1999), we propose using a no-regret learning algorithm to learn the Nash policy. The key idea originates from the self-play algorithms used in games, where the policy plays against itself to achieve self-improvement. Our contributions are summarized as follows.

**Contributions.** In this paper, we study RLHF for LLM alignment from a game-theoretic perspective. We propose a novel ***online*** algorithm called **I**terative **N**ash **P**olicy **O**ptimization (**INPO**), which learns the Nash policy of a two-player game. Our approach is built on the classical no-regret learning algorithm, online mirror descent (OMD). Unlike previous studies that also explore online algorithms for learning the Nash policy (Rosset et al., 2024; Wu et al., 2024), our approach does not require calculation of the expected win rate for each response, which is difficult to estimate accurately and may incur high costs in practice. Instead, we propose a new loss objective and prove that the minimizer of this loss uniquely corresponds to our target policy in each iteration. Therefore, similar to (Rafailov et al., 2024; Azar et al., 2024), our approach directly learns the policy over a preference dataset by minimizing the loss objective.

We prove that our algorithm approximates Nash policy with an iteration complexity of $\widetilde{\mathcal{O}}\left(\frac{1}{\epsilon^2}\right)$ and achieves last-iterate convergence at a rate of $\mathcal{O}(1/T)$. More importantly, our algorithm is easy to implement in practice, and we conduct experiments on several popular benchmarks to demonstrate its effectiveness. Remarkably, with an SFT model from LLaMA-3-8B, our INPO achieves a 42.6% length-controlled win rate on AlpacaEval 2.0 (Li et al., 2023a) and a 37.8% win rate on Arena-Hard v0.1 (Li et al., 2024), exhibiting at least 27.7% relative improvement over the state-of-the-art online RLHF algorithms (Dong et al., 2024; Wu et al., 2024).

## 2 Preliminaries

**Notations.** We use $x \in \mathcal{X}$ to denote a prompt where $\mathcal{X}$ is the prompt space. We assume that $x$ is sampled from a fixed but unknown distribution $d_0$. An LLM is characterized by a policy $\pi : \mathcal{X} \to \Delta(\mathcal{Y})$ that takes a prompt as the input and outputs a distribution over the response space $\mathcal{Y}$. A response $y \in \mathcal{Y}$ is then sampled from the distribution $\pi(\cdot|x)$. We use $\mathcal{O}(\cdot)$ to hide absolute constants and use $\widetilde{\mathcal{O}}(\cdot)$ to hide logarithmic factors. For a positive integer $T$, $[T]$ denotes the set $\{1, 2, \cdots, T\}$.

**General Preference Oracle.** We first introduce the definition of the general preference oracle as follows.

**Definition 1** (General Preference Oracle). There exists a preference oracle $\mathbb{P} : \mathcal{X} \times \mathcal{Y} \times \mathcal{Y} \to [0, 1]$, which can be queried to obtain the preference signal:

$$z \sim \text{Ber}\big(\mathbb{P}(y^1 \succ y^2 \mid x)\big),$$

where $z = 1$ means $y^1$ is preferred to $y^2$, and $z = 0$ means that $y^2$ is preferred.

Given the preference oracle, we introduce the preference distribution $\lambda_p$ (Calandriello et al., 2024). For any $x \in \mathcal{X}$ and $y, y' \in \mathcal{Y}$, we have

$$\lambda_p(x, y, y') = \begin{cases} (y, y') & \text{with probability } \mathbb{P}(y \succ y' \mid x) \\ (y', y) & \text{with probability } 1 - \mathbb{P}(y \succ y' \mid x). \end{cases} \tag{1}$$

In this paper, we study how to learn a policy $\pi$ that has a high probability of generating a preferred response over any other policy given the prompt $x$. We focus on the online setting and assume online access to the preference oracle. As demonstrated by Tang et al. (2024), online RLHF algorithms usually perform better than their offline counterparts.

## 2.1 RLHF WITH BT MODEL ASSUMPTION

**Bradley-Terry (BT) Model Assumption.** Instead of directly considering the general preference, the prevalent RLHF framework makes the Bradley-Terry (BT) model assumption. It assumes that there exists a reward function $R^*$ such that for any $x \in \mathcal{X}$ and $y^1, y^2 \in \mathcal{Y}$:

$$\mathbb{P}(y^1 \succ y^2 \mid x) = \frac{\exp(R^*(x, y^1))}{\exp(R^*(x, y^1)) + \exp(R^*(x, y^2))} = \sigma\left(R^*(x, y^1) - R^*(x, y^2)\right).$$

After learning a reward function $R$, previous RLHF algorithms aim to maximize the following KL-regularized objective:

$$J(\pi) = \mathbb{E}_{x \sim d_0}\left[\mathbb{E}_{y \sim \pi(\cdot|x)}\left[R(x, y)\right] - \tau \mathrm{KL}(\pi(\cdot|x) \| \pi_{\mathrm{ref}}(\cdot|x))\right]. \tag{2}$$

Here $\pi_{\mathrm{ref}}$ is the reference policy, which is usually a supervised fine-tuned LLM, and $\tau > 0$ is the regularization parameter. By maximizing the objective, the obtained policy simultaneously achieves a high reward and stays close to $\pi_{\mathrm{ref}}$, which can mitigate reward hacking (Tien et al., 2022; Skalse et al., 2022) to some extent.

**Direct Preference Optimization (DPO).** Rafailov et al. (2024) propose the direct preference optimization (DPO) algorithm, which directly optimizes a policy and bypasses the need to learn a reward function. The key idea is that there is a closed-form solution to Eq. (2):

$$\pi^*(y|x) \propto \pi_{\mathrm{ref}}(y|x) \exp\left(\frac{1}{\tau} R(x, y)\right),$$

which shows that each policy $\pi$ implicitly parameterizes a reward function. We can directly formulate a maximum likelihood objective to learn the optimal policy:

$$-\mathbb{E}_{x, y_w, y_l \sim \mathcal{D}}\left[\log \sigma\left(\tau \log \frac{\pi(y_w|x)}{\pi_{\mathrm{ref}}(y_w|x)} - \tau \log \frac{\pi(y_l|x)}{\pi_{\mathrm{ref}}(y_l|x)}\right)\right],$$

where $\mathcal{D}$ represents a preference dataset, $\sigma(z) = 1/(1 + \exp(-z))$ is the sigmoid function, $(y_w, y_l)$ is a preference pair for the prompt $x$, with $y_w$ being the preferred response.

## 2.2 RLHF WITH GENERAL PREFERENCES

The previously mentioned algorithms all rely on the BT model assumption, which may not hold in practice. Recently, a line of studies (Munos et al., 2023; Ye et al., 2024; Calandriello et al., 2024) directly consider the general preference $\mathbb{P}$ without additional assumptions and formulate the policy optimization problem as a two-player game. Specifically, given two policies $\pi_1$ and $\pi_2$, the game objective is written as:

$$J(\pi_1, \pi_2) = \mathbb{E}_{x \sim d_0}\left[\mathbb{E}_{y_1 \sim \pi_1, y_2 \sim \pi_2}\left[\mathbb{P}(y_1 \succ y_2 \mid x)\right] - \tau \mathrm{KL}(\pi_1(\cdot|x) \| \pi_{\mathrm{ref}}(\cdot|x)) + \tau \mathrm{KL}(\pi_2(\cdot|x) \| \pi_{\mathrm{ref}}(\cdot|x))\right], \tag{3}$$

where $\pi_1$, the max-player, aims to maximize the objective, and $\pi_2$, the min-player, aims to minimize the objective. The goal of both players is to maximize their win rates against the opponent while not deviating too far from $\pi_{\mathrm{ref}}$, which shares a similar spirit with the objective in Eq. (2).

**Nash Policy and Duality Gap.** Without loss of generality, we restrict our attention to the policy class $\Pi$ containing the policies with the same support set as $\pi_{\mathrm{ref}}$. The Nash equilibrium of the game is then defined as:

$$\pi_1^*, \pi_2^* := \underset{\pi_1 \in \Pi}{\operatorname{argmax}} \, \underset{\pi_2 \in \Pi}{\operatorname{argmin}} \, J(\pi_1, \pi_2).$$

Since the game is symmetric for the two players, as proven by Ye et al. (2024), the Nash policies of the two players are unique and coincide, meaning that $\pi_1^* = \pi_2^* = \pi^*$. We remark that for any policy $\pi \in \Pi$, we always have $J(\pi^*, \pi) \geq 0.5$, since $J(\pi^*, \pi^*) = 0.5$ and $\pi^*$ is the best response against itself. This indicates that the win rate of $\pi^*$ over any policy $\pi$ is at least one half if the KL divergence terms are negligible. Motivated by this property, our goal is to learn the Nash policy $\pi^*$. For each policy $\pi \in \Pi$, we use the following duality gap to measure how well it approximates $\pi^*$:

$$\text{DualGap}(\pi) := \max_{\pi_1 \in \Pi} J(\pi_1, \pi) - \min_{\pi_2 \in \Pi} J(\pi, \pi_2).$$

The duality gap is always non-negative and $\text{DualGap}(\pi) = 0$ only if $\pi = \pi^*$. When $\text{DualGap}(\pi) \leq \epsilon$, we say that $\pi$ is an $\epsilon$-approximate Nash policy.

## 3 ALGORITHM

In this section, we introduce our algorithm that learns the Nash policy via no-regret learning. For notation simplicity, we consider the non-contextual case and omit the prompt $x$. Since the policy processes each prompt independently, extending to the contextual case is straightforward, as shown by Azar et al. (2024).

### 3.1 ONLINE MIRROR DESCENT FOR SOLVING NASH POLICY

Given the preference oracle $\mathbb{P}$, we first consider the *__planning__* problem and introduce how to use the online mirror descent (OMD) algorithm to solve for the Nash policy. We initialize our policy $\pi_1$ as $\pi_{\text{ref}}$. At iteration $t$, our current policy is $\pi_t$ and we define the loss function for any $\pi \in \Pi$ as:

$$\ell_t(\pi) := -\mathbb{E}_{y \sim \pi, y' \sim \pi_t} \left[ \mathbb{P}(y \succ y') \right] + \tau \text{KL}(\pi \| \pi_{\text{ref}}).$$

The loss function corresponds to the game objective of the min-player with the max-player as $\pi_t$ in Eq.(3). It consists of two parts: the negative win rate of $\pi$ against current policy $\pi_t$ and the KL penalty term, which keeps $\pi$ close to the reference policy $\pi_{\text{ref}}$. A natural self-play strategy is to find $\pi_{t+1} = \text{argmin}_{\pi \in \Pi} \ell_t(\pi)$, which is the best response to $\pi_t$. However, this greedy algorithm is unstable and the next policy $\pi_{t+1}$ may deviate significantly from $\pi_t$. One can construct examples that such a greedy algorithm suffers undesirable linear regret (Lattimore & Szepesvári, 2020). Instead, in OMD with entropy regularization, also known as Hedge (Freund & Schapire, 1997), we seek the policy that minimizes the following objective:

$$\pi_{t+1} = \underset{\pi \in \Pi}{\text{argmin}} \langle \nabla \ell_t(\pi_t), \pi \rangle + \eta \text{KL}(\pi \| \pi_t), \tag{4}$$

where $\nabla_{\pi_t(y)} \ell_t(\pi_t) = -\mathbb{E}_{y' \sim \pi_t} \left[ \mathbb{P}(y \succ y') \right] + \tau \left( \log \frac{\pi_t(y)}{\pi_{\text{ref}}(y)} + 1 \right), \eta > 0$ and $\frac{1}{\eta}$ is the learning rate of OMD. Compared to the previous greedy algorithm, our objective now includes another KL divergence term between $\pi$ and $\pi_t$. The spirit is to develop a stable algorithm, requiring that the next policy $\pi_{t+1}$ not only outperforms $\pi_t$ but also stays close to $\pi_t$. Before presenting the theoretical guarantee, we make the bounded log density ratio assumption, which is also used in previous RLHF analysis (Rosset et al., 2024; Xie et al., 2024).

**Assumption A** (Bounded Log Density Ratio). For each $t \in [T]$, let $\Pi_t \subseteq \Pi$ be the feasible solution space such that $\pi_t$ obtained by OMD always belongs to $\Pi_t$. Then, for any $t \in [T]$ and $\pi \in \Pi_t$, we assume that

$$\left| \log \frac{\pi(y)}{\pi_{\text{ref}}(y)} \right| \leq B, \forall y \in \text{Supp}(\pi_{\text{ref}}).$$

In the following lemma, we show that OMD achieves sublinear regret compared to $\pi^*$. The proof directly follows from the standard analysis of the OMD algorithm (Lattimore & Szepesvári, 2020) and is deferred to Appendix A.1.

**Lemma 2** (Regret Bound for OMD). *Under Assumption A, let $D = \max_{\pi \in \Pi} \text{KL}(\pi \| \pi_1)$, OMD algorithm in Eq.* (4) *with $\eta = \frac{\max(B\tau, 1)\sqrt{T}}{\sqrt{D}}$ has the following guarantee:*

$$\sum_{t=1}^{T} \langle \nabla \ell_t(\pi_t), \pi_t \rangle - \sum_{t=1}^{T} \langle \nabla \ell_t(\pi_t), \pi^* \rangle \leq \mathcal{O}\left( \max(B\tau, 1)\sqrt{TD} \right) := \text{Reg}_T$$

We remark that in classical OMD, $\pi_1$ is a uniformly random policy and $D$ is bounded by $\log \mathcal{Y}$. Here we initialize $\pi_1$ with $\pi_{\text{ref}}$, aligning our approach with the practical RLHF workflow. With the regret bound, we are ready to show that the duality gap for uniform mixture of $\pi_t$ is well bounded.

**Theorem 3** (Duality Gap Bound for Uniform Mixture Policy in OMD). *Let* $\bar{\pi} := \frac{1}{T} \sum_{t=1}^{T} \pi_t$. *With Assumption A and* $\eta = \frac{\max(B\tau, 1)\sqrt{T}}{\sqrt{D}}$, *we have*

$$\text{DualGap}(\bar{\pi}) \leq \mathcal{O}\left( \frac{\max(B\tau, 1)\sqrt{D}}{\sqrt{T}} \right).$$

The proof mainly relies on the convexity of $\ell_t$ and Lemma 2 (see Appendix A.2). According to Theorem 3, our $\bar{\pi}$ approximates $\pi^*$ with an iteration complexity $\widetilde{\mathcal{O}}\left(\frac{1}{\epsilon^2}\right)$. Furthermore, we show that our algorithm also enjoys the last-iterate convergence to Nash policy $\pi^*$ at the speed $\mathcal{O}(1/T)$.

**Theorem 4** (Last-Iterate Convergence for OMD). *Under Assumption A, let* $C = \max(B\tau, 1)$, *at each iteration* $t$ *we have*

$$\text{KL}(\pi^*, \pi_{t+1}) \leq \left(1 - \frac{\tau}{\eta}\right) \text{KL}(\pi^*, \pi_t) + \frac{8C^2}{\eta^2}.$$

*Furthermore, suppose we use a time-varying parameter* $\eta_t = \frac{\tau(t+2)}{2}$ *in Eq. (4), we obtain*

$$\text{KL}(\pi^*, \pi_T) \leq \frac{32C^2}{\tau^2(T+1)}.$$

The proof is deferred to Appendix A.3. With Theorem 4, we can directly use the last iteration policy instead of uniformly mixing all previous policies, which makes our algorithm more practical. However, despite the OMD algorithm already enjoying a good theoretical guarantee, it assumes that we have access to $\mathbb{E}_{y \sim \pi, y' \sim \pi_t}[\mathbb{P}(y \succ y')]$ for any $\pi \in \Pi$, which is difficult to obtain in practice. Therefore, we still need to design a ***learning*** algorithm that only assumes query access to the preference oracle.

### 3.2 POPULATION LOSS

In this subsection, we introduce how to obtain a population loss objective for Eq. (4). Similar to the derivation of DPO (Rafailov et al., 2024), we start with the closed-form solution to Eq. (4):

$$\pi_{t+1}(y) \propto \pi_t(y) \exp\left(-\frac{1}{\eta} \nabla_{\pi_t(y)} \ell_t(\pi_t)\right)$$
$$\propto \exp\left(\frac{\mathbb{P}(y \succ \pi_t)}{\eta}\right) \pi_{\text{ref}}(y)^{\frac{\tau}{\eta}} \pi_t(y)^{1 - \frac{\tau}{\eta}}, \tag{5}$$

where $\mathbb{P}(y \succ \pi_t)$ represents $\mathbb{E}_{y' \sim \pi_t}[\mathbb{P}(y \succ y')]$. Note that direct computation of $\pi_{t+1}$ involves a normalization factor, which is intractable for the exponentially large response space $\mathcal{Y}$. To avoid computing this normalization factor, we consider the logarithmic ratio between response pair $y$ and $y'$, and define the function $h_t(\pi, y, y')$ as:

$$h_t(\pi, y, y') = \log \frac{\pi(y)}{\pi(y')} - \frac{\tau}{\eta} \log \frac{\pi_{\text{ref}}(y)}{\pi_{\text{ref}}(y')} - \frac{\eta - \tau}{\eta} \log \frac{\pi_t(y)}{\pi_t(y')}.$$

Unlike (Azar et al., 2024), which focuses on the offline setting and competes against $\pi_{\text{ref}}$, our algorithm operates in an online setting and iteratively competes against itself. According to the objective in Eq. (4), our target $\pi_{t+1}$ needs to stay close to both $\pi_t$ and $\pi_{\text{ref}}$ for two distinct purposes: staying close to $\pi_t$ ensures the stability of the online updates, while staying close to $\pi_{\text{ref}}$ helps avoid reward hacking. Therefore, different from its counterpart (Azar et al., 2024; Calandriello et al., 2024), which only involves $\pi_{\text{ref}}$, our $h_t$ includes both the log-likelihood of $\pi_{\text{ref}}$ and $\pi_t$. From Eq. (5), we know that the following equality holds for any response pair $y, y' \in \text{Supp}(\pi_{\text{ref}})$:

$$h_t(\pi_{t+1}, y, y') = \frac{\mathbb{P}(y \succ \pi_t) - \mathbb{P}(y' \succ \pi_t)}{\eta}. \tag{6}$$

---

**Algorithm 1** Iterative Nash Policy Optimization (INPO)

---

**Input:** Number of iterations $T$, KL regularization parameter $\tau$, OMD parameter $\eta$, reference policy $\pi_{\text{ref}}$, policy class $\Pi$, preference oracle $\mathbb{P}$.

  1: Initialize $\pi_1 \leftarrow \pi_{\text{ref}}$.
  2: **for** iteration $t = 1, 2, \ldots, T$ **do**
  3:      Use current policy $\pi_t$ to generate response pairs $\{y_1^{(i)}, y_2^{(i)}\}_{i=1}^n$ where $y_1^{(i)}, y_2^{(i)} \sim \pi_t$.
  4:      Query the preference oracle $\mathbb{P}$ to get the preference dataset $D_t = \{y_w^{(i)}, y_l^{(i)}\}_{i=1}^n$.
  5:      Calculate $\pi_{t+1}$ as:

$$\pi_{t+1} = \operatorname*{argmin}_{\pi \in \Pi} \mathbb{E}_{y_w, y_l \sim D_t} \left[ \left( h_t(\pi, y_w, y_l) - \frac{1}{2\eta} \right)^2 \right].$$

  6: **end for**
  7: Output $\pi_{T+1}$.

---

Based on this observation, we define the loss function $L_t(\pi)$ as:

$$L_t(\pi) = \mathbb{E}_{y, y' \sim \pi_t} \left[ \left( h_t(\pi, y, y') - \frac{\mathbb{P}(y \succ \pi_t) - \mathbb{P}(y' \succ \pi_t)}{\eta} \right)^2 \right]. \tag{7}$$

It is clear to see that $\pi_{t+1}$ is the minimizer of $L_t(\pi)$ since $L_t(\pi_{t+1}) = 0$. Furthermore, in the following lemma, we show that $\pi_{t+1}$ is the unique minimizer of $L_t$ within the policy class $\Pi$. The proof is deferred to Appendix A.4.

**Lemma 5.** *For each $t \in [T]$, $\pi_{t+1}$ in Eq. (5) is the unique minimizer of $L_t(\pi)$ within $\Pi$.*

Therefore, solving for $\pi_{t+1}$ is equivalent to finding a policy that minimizes $L_t(\pi)$. However, we still have the tricky term $\mathbb{P}(y \succ \pi_t)$ in our loss. To bypass this term, we propose the following population loss:

$$\mathbb{E}_{y, y' \sim \pi_t, y_w, y_l \sim \lambda_p(y, y')} \left[ \left( h_t(\pi, y_w, y_l) - \frac{1}{2\eta} \right)^2 \right]. \tag{8}$$

Recall that $\lambda_p(y, y')$ is the preference distribution defined in Eq. (1) without context. We then show the equality between $L_t(\pi)$ and Eq. (8) in the following proposition.

**Proposition 6.** *For any policy $\pi \in \Pi$ and any iteration $t \in [T]$, $L_t(\pi)$ in Eq. (7) and expression in Eq. (8) are equal up to an additive constant independent of $\pi$.*

See the proof in Appendix A.5. Here, the response pair $y, y'$ is directly sampled from the current policy $\pi_t$, which is crucial for the equivalence between $L_t(\pi)$ and Eq. (8). Additionally, this sampling is easy to implement, as we only need to perform inference using the current LLM model. In contrast, Munos et al. (2023); Calandriello et al. (2024) propose sampling from a geometric mixture between $\pi_{\text{ref}}$ and $\pi_t$, which makes implementation more challenging in practice. With the population loss in hand, we can collect a preference dataset with $\pi_t$ in each iteration and directly minimize the loss on the dataset to solve for $\pi_{t+1}$.

## 3.3 ITERATIVE NASH POLICY OPTIMIZATION ALGORITHM

We summarize our algorithm INPO in Algorithm 1. In the beginning, we initialize our policy $\pi_1$ as the reference policy $\pi_{\text{ref}}$. For each iteration $t$, we sample the current policy $\pi_t$ to generate $n$ response pairs and query the preference oracle $\mathbb{P}$ to obtain the preference dataset $D_t$. With the preference dataset, we find the policy $\pi_{t+1}$ that minimizes the sampled version of Eq. (8). Since our OMD algorithm enjoys the last-iterate convergence, we directly select the last iteration policy $\pi_{T+1}$ as our final policy, which also aligns with common practice.

### 3.4 DISCUSSION

In this subsection, we briefly discuss the differences between INPO and other general preference alignment methods, including Nash-MD (Munos et al., 2023), DNO (Rosset et al., 2024), and SPPO (Wu et al., 2024).

Nash-MD is an iterative algorithm that performs mirror descent with respect to a geometric mixture policy $\pi'_t$. However, since the response space is exponentially large, computing $\pi'_t$ exactly is intractable. Therefore, Munos et al. (2023) propose to sample from another policy that approximates $\pi'_t$. Different from Nash-MD, our INPO directly samples from the current policy $\pi_t$, which is more practical and convenient to implement. DNO first computes $\mathbb{P}(y \succ \pi_t)$ for each $y$ and then maximizes a likelihood-based learning objective. Since estimating $\mathbb{P}(y \succ \pi_t)$ accurately is challenging in practice, Rosset et al. (2024) propose a practical variant, DNO-Prct, which uses the DPO objective as an approximation. Thus, DNO-Prct can be viewed as an online version of the DPO algorithm. SPPO also incorporates $\mathbb{P}(y \succ \pi_t)$ in the update rule and they use a heuristic approximation from the dataset. In contrast, owing to the proposed loss objective in Eq. (8), INPO bypasses the computation of $\mathbb{P}(y \succ \pi_t)$ and only requires binary preference signals. This may help prevent the performance degradation caused by the estimation errors of $\mathbb{P}(y \succ \pi_t)$.

## 4 EXPERIMENTS

In this section, we use empirical results to verify the effectiveness of our INPO algorithm.

### 4.1 MAIN RESULTS

Table 1: Evaluation results on three benchmarks. RM refers to using the BT-reward model to generate preference signals, and PM refers to using the preference model to generate preference signals. The underlined results, achieved by models at least nine times larger, exceed the performance of ours.

| Model | Size | AlpacaEval 2.0 | Arena-Hard | MT-Bench |
|---|---|---|---|---|
| SFT Model | 8B | 16.0 | 10.2 | 7.52 |
| Iterative DPO (RM) | 8B | 28.3 | 24.2 | 8.22 |
| Iterative DPO (PM) | 8B | 28.5 | 29.6 | 8.29 |
| SPPO (PM) | 8B | 32.8 | 29.2 | 8.26 |
| **INPO** (RM) | 8B | **37.6** | **34.7** | **8.27** |
| **INPO** (PM) | 8B | **42.6** | **37.8** | **8.43** |
| LLaMA-3-8B-it | 8B | 24.8 | 21.2 | 7.97 |
| Tulu-2-DPO-70B | 70B | 21.2 | 15.0 | 7.89 |
| LLaMA-3-70B-it | 70B | 34.4 | 41.1 | 8.95 |
| Mixtral-8x22B-it | 141B | 30.9 | 36.4 | 8.66 |
| GPT-3.5-turbo-0613 | - | 22.7 | 24.8 | 8.39 |
| GPT-4-0613 | - | 30.2 | 37.9 | 9.18 |
| Claude-3-Opus | - | 40.5 | 60.4 | 9.00 |
| GPT-4 Turbo (04/09) | - | 55.0 | 82.6 | - |

**Settings.** We follow the online RLHF workflow (Dong et al., 2024) and begin with the same supervised fine-tuned (SFT) model[1], which is based on LLaMA-3-8B (Dubey et al., 2024), for fair comparisons. We have similar observations using other backbone models (Appendix B). The learning process of INPO lasts for $T = 3$ iterations. In each iteration, we sample responses from our current policy with a new set of prompts[2] and use preference signals on these responses to improve our policy. Instead of costly human annotations, we employ evaluation models to generate the preferences. We

---

[1] https://huggingface.co/RLHFlow/LLaMA3-SFT.
[2] Iteration 1, Iteration 2, Iteration 3.

consider two choices for evaluation models: the BT reward model[3], which is also used by Dong et al. (2024), and the preference model[4], which directly compares two responses and does not rely on the BT-model assumption. For more details on the reward model and the preference model, please refer to (Dong et al., 2024).

We follow the rejection sampling strategy suggested by Dong et al. (2024). For each prompt, we generate $K = 8$ responses and use the best-of-8 as $y_w$ and the worst-of-8 as $y_l$. For the BT reward model, we directly select the response with the highest reward as the best and the response with the lowest reward as the worst. For the preference model, we use a tournament approach, selecting the winner as the best and the loser as the worst. We first split eight samples into four pairs and compare each pair. If the result is a tie, we select the first one as the winner. Then, the winners are compared against each other and the losers against each other until we get the final winning response $y_w$ and losing response $y_l$. We finally compare $y_w$ with $y_l$ and only train the model with the pairs where $y_w$ wins over $y_l$. We need eleven comparisons in total for eight responses. We remark that compared to (Wu et al., 2024), which estimates the expected win rate and requires $\mathcal{O}(K^2)$ preference queries, our tournament strategy only needs $\mathcal{O}(K)$ queries.

We evaluate the model performance on three widely used benchmarks: MT-Bench (Zheng et al., 2024), AlpacaEval 2.0 (Li et al., 2023a), and Arena-Hard v0.1 (Li et al., 2024). MT-Bench contains 80 questions from eight categories, with answers rated by GPT-4 on a scale of 1-10. Arena-Hard v0.1 contains 500 technical problem-solving questions, and the answers are compared to reference responses from the baseline model GPT-4-0314. We report the win rate (WR) as judged by GPT-4 Turbo (Preview-1106). AlpacaEval 2.0 includes 805 questions from five datasets, with the judge model GPT-4 Turbo (Preview-1106) comparing the answers to reference responses from itself. We report the length-controlled (LC) WR as suggested by Dubois et al. (2024).

**Results and Analysis.** We compare our INPO with the state-of-the-art online alignment methods, including iterative DPO (Dong et al., 2024) and SPPO (Wu et al., 2024) (see implementation details in Appendix B), as shown in Table 1. Note that SPPO algorithm requires the score from a pair preference model. Therefore, it is only implemented with the preference model (PM). We observe that INPO outperforms baselines on all three benchmarks, with notable improvements on AlpacaEval 2.0 and Arena-Hard v0.1. Additionally, we compare INPO with other open-source and closed-source LLMs, including LLaMA-3-70B-it, GPT-4-0613, Claude-3-Opus, and GPT-4 Turbo (numbers copied from (Dong et al., 2024)). For AlpacaEval 2.0, our INPO is only surpassed by GPT-4 Turbo and outperforms all other models. According to the results in (Dubois et al., 2024), LC AlpacaEval 2.0 has the highest correlation with Chatbot Arena (Zheng et al., 2024), highlighting the superior performance achieved by INPO.

Moreover, we note that methods utilizing the preference model as the oracle generally outperform those relying on the BT reward model as the oracle. This observation aligns with the results from previous studies (Ye et al., 2024; Dong et al., 2024), which show that the preference model outperforms the BT reward model on RewardBench (Lambert et al., 2024), demonstrating the importance of considering general preferences without the BT model assumption.

## 4.2 Results on More Academic Benchmarks

Table 2: Model performance on more academic benchmarks (AVG: average).

| Model | IFEval | GPQA | MMLU | Hellaswag | TruthfulQA | GSM8K | AVG |
|---|---|---|---|---|---|---|---|
| SFT Model | 35.2 | 30.2 | 62.4 | 78.6 | 53.4 | 73.4 | 55.5 |
| Iterative DPO | 37.3 | 29.8 | 63.1 | 80.5 | 60.7 | 81.3 | 58.8 |
| SPPO | 40.4 | 29.0 | 63.1 | 80.8 | 63.0 | 80.9 | 59.5 |
| INPO | 41.6 | 28.9 | 63.1 | 80.8 | 64.9 | 80.8 | **60.0** |

It is known that RLHF alignment may have a negative effect on a model's abilities in reasoning, calibration, and generating accurate responses (Ouyang et al., 2022; Bai et al., 2022c; Dong et al.,

---

[3] https://huggingface.co/sfairXC/FsfairX-LLaMA3-RM-v0.1.
[4] https://huggingface.co/RLHFlow/pair-preference-model-LLaMA3-8B.

2024). Therefore, it is necessary to evaluate the model performance on more academic benchmarks. In this subsection, we present the results on six benchmarks, evaluating various model abilities including explicit instruction following (Zhou et al., 2023), general knowledge (Rein et al., 2023), multitask language understanding (Hendrycks et al., 2020), commonsense reasoning (Zellers et al., 2019), human falsehoods mimicking (Lin et al., 2021), and math word problem-solving (Cobbe et al., 2021). We compare our INPO (PM) with the SFT baseline, iterative DPO (PM), and SPPO (PM). The results are shown in Table 2.

Interestingly, compared to the SFT baseline, all three alignment methods exhibit performance improvements on these benchmarks. A potential reason for this is that during the alignment stage, the alignment methods more effectively leverage the model's internal knowledge and abilities, which were introduced during the pre-training and SFT stages. Additionally, both INPO and iterative DPO incorporate KL regularization, which prevents the learned policy from deviating significantly from the reference policy, thereby avoiding performance degradation. And the superior results of INPO and SPPO demonstrate the advantage of considering general preferences.

### 4.3 ABLATION STUDIES OF KL REGULARIZATION

Table 3: Ablation study of KL regularization term. For INPO w/o KL, we set $\tau$ to be zero in $h_t(\pi, y, y')$.

| Preference Oracle | Model | AlpacaEval 2.0 | Arena-Hard v0.1 | MT-Bench |
|---|---|---|---|---|
| BT Reward Model | INPO w/o KL | 35.4 | 33.6 | 8.10 |
| | INPO w/ KL | **37.6** | **34.7** | **8.27** |
| Preference Model | INPO w/o KL | 41.6 | 36.5 | 8.31 |
| | INPO w/ KL | **42.6** | **37.8** | **8.43** |

In this subsection, we conduct an ablation study to examine the benefits of including the KL regularization term in the game objective. The results are shown in Table 3. We observe that INPO with KL regularization (INPO w/ KL) generally outperforms its counterpart without KL regularization (INPO w/o KL) by a clear margin. This indicates regularizing our policy towards the reference policy is beneficial for the alignment performance.

## 5 RELATED WORK

**Reward-Based RLHF.** Since RLHF has achieved great success in LLM alignment (Ouyang et al., 2022; Touvron et al., 2023; Achiam et al., 2023), it has been extensively studied, including using RL algorithms such as PPO (Schulman et al., 2017) to maximize a KL-regularized objective (Bai et al., 2022c; Korbak et al., 2022; Li et al., 2023b) and reward-ranked finetuning (Dong et al., 2023; Yuan et al., 2023; Gulcehre et al., 2023). Recently, Rafailov et al. (2024) propose the DPO algorithm, which directly optimizes the policy on a preference dataset, bypassing the need for reward model training. Further studies by Xiong et al. (2024); Dong et al. (2024); Xie et al. (2024) investigate the online variant of DPO, proposing iterative algorithms with different exploration strategies. However, all these methods are reward-based and rely on the BT model assumption. In this paper, we study RLHF from a game-theoretic perspective and consider general preferences.

**RLHF under General Preferences.** (Azar et al., 2024) is the first work to consider general preferences, proposing an offline algorithm IPO that learns the best policy against the reference policy. Munos et al. (2023) formulate LLM alignment as a two-player game and propose a planning algorithm to solve for the Nash policy when the general preference oracle is given. Ye et al. (2024) provide theoretical analysis for both offline and online algorithms that learn the Nash policy in the game. Calandriello et al. (2024) propose the online IPO algorithm and prove that the minimizer of the online IPO objective is the Nash policy of the game. However, their algorithm uses the policy gradient method, and the effective minimization of the objective remains unclear. Rosset et al. (2024) propose an iterative algorithm to learn the Nash policy, they assume that the learner has access to the expected win rate of each response, which serves a similar role to the reward of the response.

The closest related work to ours is (Wu et al., 2024), which also uses no-regret learning algorithms. However, they study the game without KL-regularized terms. More importantly, their algorithm still requires the estimation of the expected win rate, leading to square oracle query complexity that may incur high costs in practice. Instead, our algorithm directly optimizes the policy over a preference dataset and bypasses the need for win rate estimation.

**No-Regret Learning in Games.** There has been a long history of using no-regret learning to solve for the equilibrium of games, including matrix games (Freund & Schapire, 1999; Daskalakis et al., 2011; Rakhlin & Sridharan, 2013; Syrgkanis et al., 2015; Chen & Peng, 2020; Wei et al., 2020; Daskalakis et al., 2021; Zhang et al., 2022), extensive-form games (Kozuno et al., 2021; Bai et al., 2022a;b; Fiegel et al., 2023) and Markov games (Bai et al., 2020; Song et al., 2021; Jin et al., 2021; Mao & Başar, 2023). Our problem formulation can be viewed as a contextual case of the two-player matrix game, and we use the classical OMD algorithm to learn the Nash equilibrium.

## 6 CONCLUSION AND FUTURE WORK

In this work, we consider RLHF under general preferences and formulate it as a two-player game. Building on no-regret learning, we propose a new online algorithm, iterative Nash policy optimization (INPO), to learn the Nash policy of the game. To bypass the estimation of the expected win rate, we design a new loss objective, and our algorithm directly minimizes it over a preference dataset. Our INPO algorithm not only has good theoretical guarantees but also empirically outperforms state-of-the-art online RLHF algorithms across various benchmarks. In the future, we plan to study the finite-sample analysis of our algorithm and extend it to the general reinforcement learning framework, such as Markov decision processes.

## ACKNOWLEDGMENTS

Nan Jiang acknowledges funding support from NSF CNS-2112471, NSF CAREER IIS-2141781, Google Scholar Award, and Sloan Fellowship.

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

# A PROOFS FOR SECTION 3

## A.1 PROOF FOR LEMMA 2

*Proof.* According to the classical analysis of OMD algorithm (Lattimore & Szepesvári, 2020), for any policy $\pi$, we have

$$\sum_{t=1}^{T} \langle \nabla \ell_t(\pi_t), \pi_t \rangle - \sum_{t=1}^{T} \langle \nabla \ell_t(\pi_t), \pi \rangle \leq \eta \mathrm{KL}(\pi \| \pi_1) + \frac{1}{\eta} \sum_{t=1}^{T} \| \nabla \ell_t(\pi_t) \|_{\infty}^2$$

$$\leq \eta D + \frac{(4\tau^2 B^2 + 1)T}{\eta}.$$

In the second step, w.l.o.g., we assume $B \geq 1$. Picking $\eta = \frac{\max(B\tau, 1)\sqrt{T}}{\sqrt{D}}$ finishes the proof. $\quad\square$

## A.2 PROOF FOR THEOREM 3

*Proof.* We first decompose $\mathrm{DualGap}(\bar{\pi})$ as

$$\mathrm{DualGap}(\bar{\pi}) = \underbrace{\max_{\pi_1} J(\pi_1, \bar{\pi}) - J(\pi^*, \pi^*)}_{\text{Term A}} + \underbrace{J(\pi^*, \pi^*) - \min_{\pi_2} J(\bar{\pi}, \pi_2)}_{\text{Term B}}.$$

Next, we show how to bound Term A. Since $\ell_t$ is convex for all $t$, for any $\pi$, we have

$$\sum_{t=1}^{T} \ell_t(\pi_t) - \sum_{t=1}^{T} \ell_t(\pi) \leq \sum_{t=1}^{T} \langle \nabla \ell_t(\pi_t), \pi_t \rangle - \sum_{t=1}^{T} \langle \nabla \ell_t(\pi_t), \pi \rangle \leq \mathrm{Reg}_T. \tag{9}$$

According to the definition of $\ell_t$, we also get that

$$\frac{1}{T} \sum_{t=1}^{T} (\ell_t(\pi_t) - \ell_t(\pi))$$

$$= \frac{1}{T} \sum_{t=1}^{T} \left( -\mathbb{E}_{y \sim \pi_t, y' \sim \pi_t} [\mathbb{P}(y \succ y')] + \tau \mathrm{KL}(\pi_t \| \pi_{\mathrm{ref}}) + \mathbb{E}_{y \sim \pi, y' \sim \pi_t} [\mathbb{P}(y \succ y')] - \tau \mathrm{KL}(\pi \| \pi_{\mathrm{ref}}) \right)$$

$$= \frac{1}{T} \sum_{t=1}^{T} \left( \mathbb{E}_{y \sim \pi, y' \sim \pi_t} [\mathbb{P}(y \succ y')] + \tau \mathrm{KL}(\pi_t \| \pi_{\mathrm{ref}}) \right) - \tau \mathrm{KL}(\pi \| \pi_{\mathrm{ref}}) - \frac{1}{2}$$

$$\geq J(\pi, \bar{\pi}) - \frac{1}{2} = J(\pi, \bar{\pi}) - J(\pi^*, \pi^*). \tag{10}$$

The inequality is from Jensen's inequality and convexity of KL divergence. Combining Eq. (9) and Eq. (10), we obtain that for any $\pi$

$$J(\pi, \bar{\pi}) - J(\pi^*, \pi^*) \leq \frac{\mathrm{Reg}_T}{T}.$$

Since the game is symmetric, Term B can also be bounded similarly. Finally, we get

$$\mathrm{DualGap}(\bar{\pi}) \leq \frac{2\mathrm{Reg}_T}{T} \leq \mathcal{O}\left( \frac{\max(B\tau, 1)\sqrt{D}}{\sqrt{T}} \right).$$

The proof is completed. $\quad\square$

## A.3 PROOF FOR THEOREM 4

We start with a useful lemma for OMD.

**Lemma 7** (Lemma 2 in Munos et al. (2023)). *Let $p \geq 1$ and $q \geq 1$ such that $1/p + 1/q = 1$. Let $\phi$ be a $\sigma$-strongly convex function with respect to the $\ell_p$-norm $\| \cdot \|_p$, i.e., for any $\pi, \pi'$,*

$$\phi(\pi) \geq \phi(\pi') + \nabla \phi(\pi') \cdot (\pi - \pi') + \frac{\sigma}{2} \| \pi - \pi' \|^2.$$

*Let $D_\phi$ be the associated Bregman divergence: for $\pi, \pi'$,*

$$D_\phi(\pi, \pi') := \phi(\pi) - \phi(\pi') - \nabla\phi(\pi') \cdot (\pi - \pi').$$

*Let $\delta$ be a vector of dimension $|\mathcal{Y}|$. For any $\pi^- \in \Delta(\mathcal{Y})$, define $\pi^+$ as*

$$\pi^+ = \arg \max_{\pi \in \Delta(\mathcal{Y})} \left[ \sum_y \pi(y)\delta(y) - D_\phi(\pi, \pi^-) \right],$$

*Then for any $\pi \in \Delta(\mathcal{Y})$, we have,*

$$D_\phi(\pi, \pi^+) \le D_\phi(\pi, \pi^-) + \sum_y (\pi^-(y) - \pi(y))\delta(y) + (2/\sigma)\|\delta\|_q^2.$$

We then prove Theorem 4.

*Proof.* We invoke Lemma 7 with $\pi^- = \pi_t$, $\pi^+ = \pi_{t+1}$, $\phi(\pi) = \sum_y \pi(y) \log \pi(y)$ and $\delta(y) = \frac{1}{\eta}\mathbb{P}(y \succ \pi_t) - \frac{\tau}{\eta}\left(\log \frac{\pi_t(y)}{\pi_{\text{ref}}(y)} + 1\right)$. For notation simplicity, we use $\mathbb{P}(\pi_1 \succ \pi_2)$ to represent $\mathbb{E}_{y \sim \pi_1, y' \sim \pi_2}[\mathbb{P}(y \succ y')]$. Then, at iteration $t$, we get

$$\text{KL}(\pi^*, \pi_{t+1})$$

$$\le \text{KL}(\pi^*, \pi_t) + \frac{1}{\eta} \sum_y (\pi_t(y) - \pi^*(y)) \left( \mathbb{P}(y \succ \pi_t) - \tau \log \frac{\pi_t(y)}{\pi_{\text{ref}}(y)} \right) + 2\|\delta\|_\infty^2$$

$$\le \left(1 - \frac{\tau}{\eta}\right) \text{KL}(\pi^*, \pi_t) + \frac{1}{\eta}\left(\frac{1}{2} - \tau\text{KL}(\pi_t, \pi_{\text{ref}}) - \mathbb{P}(\pi^* \succ \pi_t)\right) + \frac{\tau}{\eta}\sum_y \pi^*(y)\left(\log\frac{\pi^*(y)}{\pi_t(y)} + \log\frac{\pi_t(y)}{\pi_{\text{ref}}(y)}\right) + 2\|\delta\|_\infty^2$$

$$\le \left(1 - \frac{\tau}{\eta}\right) \text{KL}(\pi^*, \pi_t) + \frac{1}{\eta}\left(\frac{1}{2} - \tau\text{KL}(\pi_t, \pi_{\text{ref}}) - \mathbb{P}(\pi^* \succ \pi_t) + \tau\text{KL}(\pi^*, \pi_{\text{ref}})\right) + 2\|\delta\|_\infty^2$$

$$\le \left(1 - \frac{\tau}{\eta}\right) \text{KL}(\pi^*, \pi_t) + 2\|\delta\|_\infty^2.$$

The last step is because $\pi^*$ is the Nash policy and $J(\pi^*, \pi^*) = \frac{1}{2}$. W.l.o.g., we assume $B \ge 1$ and have

$$\|\delta\|_\infty = \frac{1}{\eta} \left\| -\mathbb{P}(y \succ \pi_t) + \tau\left(\log\frac{\pi_t(y)}{\pi_{\text{ref}}(y)} + 1\right) \right\|_\infty \le \frac{2C}{\eta}.$$

Now, we obtain

$$\text{KL}(\pi^*, \pi_{t+1}) \le \left(1 - \frac{\tau}{\eta}\right) \text{KL}(\pi^*, \pi_t) + \frac{8C^2}{\eta^2}.$$

Suppose we use time-varying $\eta_t = \frac{\tau(t+2)}{2}$, when $t = 0$, $\eta_0 = \tau$, and we have

$$\text{KL}(\pi^*, \pi_1) \le \frac{8C^2}{\tau^2}.$$

By induction, assuming $\text{KL}(\pi^*, \pi_t) \le \frac{32C^2}{\tau^2(t+1)}$, we further get

$$\text{KL}(\pi^*, \pi_{t+1}) \le \left(1 - \frac{2}{t+2}\right)\frac{32C^2}{\tau^2(t+1)} + \frac{32C^2}{\tau^2(t+2)^2}$$

$$\le \left(1 - \frac{2}{t+2} + \frac{1}{t+2}\right)\frac{32C^2}{\tau^2(t+1)}$$

$$\le \frac{32C^2}{\tau^2(t+2)}.$$

The proof is completed. $\qquad\square$

## A.4 Proof for Lemma 5

*Proof.* We use contradiction to prove the lemma. Let $\widetilde{\pi} \in \Pi$ be another policy such that $\widetilde{\pi} \neq \pi_{t+1}$ and $L_t(\widetilde{\pi}) = 0$. Let $y$ be an arbitrary element from $\mathcal{Y}$. For any other $y' \in \mathrm{Supp}(\pi_{\mathrm{ref}})$ and $y' \neq y$, we have

$$\frac{\widetilde{\pi}(y)}{\widetilde{\pi}(y')} = \frac{\exp\left(\frac{\mathbb{P}(y \succ \pi_t)}{\eta}\right) \pi_{\mathrm{ref}}(y)^{\frac{\tau}{\eta}} \pi_t(y)^{1-\frac{\tau}{\eta}}}{\exp\left(\frac{\mathbb{P}(y' \succ \pi_t)}{\eta}\right) \pi_{\mathrm{ref}}(y')^{\frac{\tau}{\eta}} \pi_t(y')^{1-\frac{\tau}{\eta}}}. \tag{11}$$

Since $\mathrm{Supp}(\widetilde{\pi}) = \mathrm{Supp}(\pi_{\mathrm{ref}})$, we also have $\sum_{y' \in \mathrm{Supp}(\pi_{\mathrm{ref}})} \widetilde{\pi}(y') = 1$. Hence, the value of $\widetilde{\pi}(y)$ is uniquely determined. Because $\pi_{t+1}$ also satisfies Eq. 11 and shares the same support set as $\widetilde{\pi}$, we have $\widetilde{\pi}(y) = \pi_{t+1}(y)$ and hence $\widetilde{\pi}(y') = \pi_{t+1}(y')$ for all $y' \in \mathcal{Y}$, contradicting with $\widetilde{\pi} \neq \pi_{t+1}$. Therefore, the minimizer is unique and the proof is completed. $\square$

## A.5 Proof for Proposition 6

*Proof.* We first consider the following expression and show that it equals to $L_t(\pi)$ up to some constants:

$$\mathbb{E}_{y,y' \sim \pi_t, I \sim \mathrm{Ber}(\mathbb{P}(y \succ y'))} \left[ \left( h_t(\pi, y, y') - \frac{I}{\eta} \right)^2 \right]. \tag{12}$$

It suffices to show that

$$\mathbb{E}_{y,y'} \left[ h_t(\pi, y, y')(\mathbb{P}(y \succ \pi_t) - \mathbb{P}(y' \succ \pi_t)) \right] = \mathbb{E}_{y,y',I} \left[ h_t(\pi, y, y')I \right].$$

Let $p_y = \mathbb{P}(y \succ \pi_t)$ and $\pi_y = \log \pi(y)$, $\pi_{\mathrm{ref},y} = \frac{\tau}{\eta} \log \pi_{\mathrm{ref}}(y)$ and $\pi_{t,y} = (1 - \frac{\tau}{\eta}) \log \pi_t(y)$. For RHS, it can be written as

$$\mathbb{E}_{y,y',I} \left[ h_t(\pi, y, y')I \right]$$
$$= \mathbb{E}_{y,y',I} \left[ (\pi_y - \pi_{y'} - \pi_{\mathrm{ref},y} + \pi_{\mathrm{ref},y'} - \pi_{t,y} + \pi_{t,y'}) I \right]$$
$$= \mathbb{E}_y \left[ (\pi_y - \pi_{\mathrm{ref},y} - \pi_{t,y}) \mathbb{E}_{y',I}[I] \right] + \mathbb{E}_{y'} \left[ (-\pi_{y'} + \pi_{\mathrm{ref},y'} + \pi_{t,y'}) \mathbb{E}_{y,I}[I] \right]$$
$$= \mathbb{E}_{y,y'} \left[ \pi_y p_y - \pi_{\mathrm{ref},y} p_y - \pi_{t,y} p_y - (1 - p_{y'}) \pi_{y'} + (1 - p_{y'}) \pi_{\mathrm{ref},y'} + (1 - p_{y'}) \pi_{t,y'} \right]$$
$$= \mathbb{E}_y \left[ (2p_y - 1)\pi_y - (2p_y - 1)\pi_{\mathrm{ref},y} - (2p_y - 1)\pi_{t,y} \right].$$

In the last step, we use the fact that $y$ and $y'$ are from the same distribution. The LHS can be written as

$$\mathbb{E}_{y,y'} \left[ h_t(\pi, y, y')(\mathbb{P}(y \succ \pi_t) - \mathbb{P}(y' \succ \pi_t)) \right]$$
$$= \mathbb{E}_{y,y'} \left[ (\pi_y - \pi_{y'} - \pi_{\mathrm{ref},y} + \pi_{\mathrm{ref},y'} - \pi_{t,y} + \pi_{t,y'}) (p_y - p_{y'}) \right]$$
$$= \mathbb{E}_{y,y'} \left[ 2p_y \pi_y - p_y \pi_{y'} - p_{y'} \pi_y - 2p_y \pi_{\mathrm{ref},y} + p_{y'} \pi_{\mathrm{ref},y} + p_y \pi_{\mathrm{ref},y'} - 2p_y \pi_{t,y} + p_{y'} \pi_{t,y} + p_y \pi_{t,y'} \right]$$
$$= \mathbb{E}_y \left[ (2p_y - 1)\pi_y - (2p_y - 1)\pi_{\mathrm{ref},y} - (2p_y - 1)\pi_{t,y} \right].$$

The second equality is from that $y$ and $y'$ are from the same distribution. The last equality is from that $\mathbb{E}_y[p_y] = \frac{1}{2}$. Therefore, we show the equivalence between $L_t(\pi)$ and Eq. 12. Next, we show the equivalence between Eq. 8 and Eq. 12. We expand the expectation over $\lambda_p(y, y')$ and rewrite Eq. 8 as

$$\mathbb{E}_{y,y'} \left[ \mathbb{P}(y \succ y') \left( h_t(\pi, y, y') - \frac{1}{2\eta} \right)^2 + (1 - \mathbb{P}(y \succ y')) \left( h_t(\pi, y', y) - \frac{1}{2\eta} \right)^2 \right].$$

We also expand the expectation over $I$ in Eq. 12 and write it as

$$\mathbb{E}_{y,y'} \left[ \mathbb{P}(y \succ y') \left( h_t(\pi, y, y') - \frac{1}{\eta} \right)^2 + (1 - \mathbb{P}(y \succ y')) h_t(\pi, y, y')^2 \right].$$

Ignoring the constants, since $h_t(\pi, y, y') = -h_t(\pi, y', y)$, the difference is:

$$\frac{1}{\eta} \mathbb{E}_{y,y'} \left[ \mathbb{P}(y \succ y') h_t(\pi, y, y') - (1 - \mathbb{P}(y \succ y')) h_t(\pi, y', y) \right]. \tag{13}$$

For each pair $y, y'$, it will appear two times in the expectation and the total contribution is:

$$\frac{\pi_t(y)\pi_t(y')}{\eta} \left( \mathbb{P}(y \succ y') h_t(\pi, y, y') - \mathbb{P}(y' \succ y) h_t(\pi, y', y) + \mathbb{P}(y' \succ y) h_t(\pi, y', y) - \mathbb{P}(y \succ y') h_t(\pi, y, y') \right) = 0.$$

Therefore, the expression in Eq. (13) equals to zero and the proof is completed. $\square$

## B ADDITIONAL EXPERIMENT DETAILS AND RESULTS

**Implementation Details.** We implement iterative DPO according to Dong et al. (2024) and their GitHub repository [5]. We implement SPPO according to the official Github repository [6]. For the implementation of INPO, we follow the hyperparameters in Dong et al. (2024), including the cosine learning rate scheduler with a peak learning rate of $5 \times 10^{-7}$, a 0.03 warm-up ratio, and a global batch size of 128. We use a grid search for $\eta$ over $[0.1, 0.01, 0.0075, 0.005, 0.002]$ and set $\eta = 0.005$. $\tau$ is directly set to be one-third of $\eta$.

**Additional Experiment Results.** In the main text, we use a SFT model from LLaMA-3-8B as our base model. Here, we also conduct experiments with Llama-3-8B-Instruct[7], an instruction tuned model. The results on three alignment benchmarks and six academic benchmarks are presented in Table 4 and Table 5, respectively. As shown in the results, our INPO consistently outperforms the baselines. However, the improvement is less significant than when using the SFT model as the starting point. This is likely because the instruct model has already been fine-tuned using RLHF methods, which may limit the potential for further improvement through additional training. Therefore, fine-tuning starting from the SFT model may offer a greater scope for enhancement.

Table 4: Results on three alignment benchmarks using LLaMA-3-8B-It as the base model.

| Model | AlpacaEval 2.0 | Arena-Hard | MT-Bench |
|---|---|---|---|
| LLaMA-3-8B-It | 24.8 | 21.2 | 7.97 |
| Iterative DPO | 35.4 | 37.1 | 8.35 |
| SPPO | 39.2 | 37.9 | 8.42 |
| INPO | **41.8** | **42.5** | **8.43** |

Table 5: Results on six academic benchmarks using LLaMA-3-8B-It as the base model.

| Model | IFEval | GPQA | MMLU | Hellaswag | TruthfulQA | GSM8K | Average |
|---|---|---|---|---|---|---|---|
| LLaMA-3-8B-It | 47.6 | 31.4 | 63.9 | 75.8 | 51.7 | 76.4 | 57.8 |
| Iterative DPO | 41.5 | 30.8 | 64.2 | 76.3 | 55.9 | 74.2 | 57.2 |
| SPPO | 43.0 | 30.7 | 64.1 | 75.0 | 57.2 | 74.8 | 57.5 |
| INPO | 42.6 | 31.0 | 64.0 | 75.3 | 57.9 | 76.8 | **57.9** |

---

[5] https://github.com/RLHFlow/Online-RLHF.
[6] https://github.com/uclaml/SPPO.
[7] https://huggingface.co/meta-llama/Meta-Llama-3-8B-Instruct.

