# OpenReview forum: "Iterative Nash Policy Optimization: Aligning LLMs with General Preferences via No-Regret Learning"
_ICLR.cc/2025/Conference — ICLR 2025 Oral_

### Official Review · Reviewer_CnZh · 2024-10-25

**Soundness:** 4
**Presentation:** 2
**Contribution:** 3
**Rating:** 6
**Confidence:** 2

**Summary:**

This work proposes an RLHF algorithm, extending SPPO by Wu et al. (2024) to an algorithm that does not rely a score function, but directly optimizes over the preferences data, similarly to DPO.

This algorithm, motivated by a Nash equilibrium formulation and no regret approaches to computing Nash equilibria, is compared on multiple benchmarks with state of the art algorithms.

**Strengths:**

This work presents extensive experiments, comparing the proposed algorithm with multiple state-of-the-art algorithms, on many numerous benchmarks.

The proposed algorithm INPO seems promising: it does not rely on a prior score function and outperforms (on average) other algorithms on multiple benchmarks. Moreover, the iNPO algorithm is quite simple, allowing an easy implementation.

**Weaknesses:**

At the reading of the paper, it seems hard to really understand what contributions are due to this work and how it compares with previous works.

Moreover, it is not clear at first sight why optimizing directly on the preferences data (without using a score function) is an advantage. When looking at table 2, INPO seems to perform similarly to SPPO, so I believe an extended discussion of the advantage of INPO would be required. Lines 462-465, the authors seem to suggest that the benefit is mostly from a computational point of view. In that case, I believe a comparison of the computation cost for running the different algorithms would also be needed.

Lastly, all the tables are given without any confidence interval or standard error. As a consequence, it is hard to really assess the superiority of INPO wrt other algorithms.

As a minor remark, lines 154-155: the phrasing seem to suggest that the uniqueness of equilibrium is due to the symmetry of the game. That's not the case, the uniqueness is actually due to the strict concavity/convexity of the players' utilities.

**Questions:**

- While the algorithmic difference between INPO and SPPO is clearly established in the paper, how does it impact the practice?
- Could you provide uncertainty quantifications in the different tables of section 4?

---

### Official Review · Reviewer_jJq8 · 2024-10-31

**Soundness:** 2
**Presentation:** 3
**Contribution:** 2
**Rating:** 6
**Confidence:** 4

**Summary:**

The authors adopt a game-theoretic approach to study RLHF, where the problem is framed as a two-player game and is solved by an online algorithm called Iterative Nash Policy Optimization (INPO). The main idea is to allow the policy to compete against itself using no-regret learning, which approximates a Nash equilibrium policy.

**Strengths:**

1. Theoretical Framework: The paper models RLHF as a two-player game, extending traditional RLHF approaches beyond the commonly used Bradley-Terry (BT) model for estimating preferences. This approach allows for a more nuanced understanding of preferences that may not be transitive or linear, which is a significant step forward in preference modeling.
2. No-Regret Learning: The use of no-regret learning algorithms, particularly Online Mirror Descent (OMD), ensures theoretical soundness and guarantees convergence to a Nash equilibrium.
3. Empirical Validation: Demonstrating empirical improvements over existing state-of-the-art RLHF algorithms on several benchmarks, the model's effectiveness is validated through comprehensive experimental results.

**Weaknesses:**

1. Assumption Sensitivity: The performance of the algorithm heavily relies on the assumptions made about the preference oracle's availability and the nature of the loss functions used. Any deviation in the real-world application from these assumptions could significantly impact the model's performance.
2. Loss Function: The loss function used in INPO is quite similar to that used in the IPO, which may not offer significant improvements in differentiating between the complexities or subtleties of human preferences beyond what IPO already provides.
3. Computational Complexity: Although the paper claims reduced computational complexity due to direct loss minimization over a preference dataset, this assertion does not fully account for the computational demands that might arise when solving the No-regret learning optimization problem.
4. Limited Experiments: The experimental validation is restricted to the LLaMA-3-8B model, which has been specifically tuned through supervised fine-tuning (SFT) and RLHF for alignment with human preferences concerning helpfulness and safety.

**Questions:**

1. Can the approach be extended to multi-player games beyond the two-player settings considered in the paper?
2. What are the implications of the bounded log density ratio assumption in practical scenarios, and how sensitive is INPO to violations of this assumption?
3. How sensitive is OMD to the choice of loss function, especially considering different types of loss functions that might be used to capture complex human preferences? Can minor changes in the loss function formulation lead to significantly different outcomes?
4. What mechanisms are in place to ensure that the preference oracle does not introduce biases that could skew the learning process?
5. How does INPO compare with this work: Remi Munos et al., Nash Learning from Human Feedback, https://arxiv.org/pdf/2312.00886, since they share the same concept of Nash learning for aligning LLMs?

---

### Official Review · Reviewer_1fVL · 2024-10-31

**Soundness:** 3
**Presentation:** 3
**Contribution:** 3
**Rating:** 6
**Confidence:** 4

**Summary:**

RLHF involves optimizing a policy to align a language model with human preferences. It is generally done by separately training a reward model and then uses RL steps with this reward model as feedback as well as a regularization to ensure that the model does not drift too far from the initially trained LM.

The paper proposes an algorithm to bypass the training of a separate reward model (this line of work has been introduced by the Direct Preference Optimization algorithm, Rafailov et al., 2024). It does so by iteratively optimizing the policy in an online learning fashion. Similarly to Nash Learning from Human Feedback, Munos et al. (2023), it formulates the problem as a two-players game where the Nash equilibrium is the symmetric optimal policy for both players. The duality gap represents how close from the optimum $\pi*$ the policies are.

Then, the paper introduces the iterative Nash policy optimization (INPO) algorithm to solve this optimization problem. Basically, INPO is an online mirror descent tailored for this specific problem. Theorems 3 and 4 prove convergence (Theorem 3 gives a bound on the duality gap for the averaged policy over $T$ rounds that scales in $O(1/ \sqrt{T})$). Several experiments support the paper.

**Strengths:**

Even though it already exists, I appreciate the formulation of RLHF as a game and the leveraging of Online Mirror Descent to optimize it since there seems to be a promising intersection between game theory and machine learning.

The claims are supported by extensive experiments, which make it relevant for ICLR. Theoretical proofs are also given.

I must admit that the paper is very clear and easy to follow, which is quite pleasant.

**Weaknesses:**

It is very nice to provide convergence guarantees for RLHF algorithms but my main concern is about their novelty. Theorem 3 is nothing new as compared to what already exists in proofs of convergence for OMD in the literature. Lemma 7 is presented as lemma 2 in Munos et al. (2023) but it has already been introduced in Munos et al. (2020) (Fast computation of Nash Equilibria in Imperfect Information Games). Theorem 4 is only three lines of algebra once Lemma 7 is given.

More generally, I wonder what are the paper's contributions, except slight variations around existing work in RLHF, Online Learning Theory and some previous papers (e.g., Munos et al., 2023, Rosset et al., 2024).

Assumption A seems quite strong and deserves at least more discussion.

**Questions:**

A lot of work prove the interest of online versus offline RLHF algorithms and some works even propose online learning for Nash optimization, such as “Direct Nash Optimization: Teaching Language Models to Self-Improve with General Preferences”, that you cite. It is mentionned that there is no need to compute the win-rate in INPO as compared to the latter. However, except a trick to bypass this, I do not see to what extent INPO really brings something new as compared to “Direct Nash Optimization…”. At least, a deeper comparison would be needed. Except slight variations, is it possible to have a clearer description of the true contributions of the paper?

It is mentioned that INPO is better than its offline counterparts, a claim that is quite common in the RLHF literature. However, is there a guarantee to bound the convergence rates or the precision of an online versus an offline procedure here? Or at least, is it possible to have more details about what is meant by being better than the offline counterpart?

---

### Official Review · Reviewer_2vvo · 2024-10-31

**Soundness:** 3
**Presentation:** 3
**Contribution:** 3
**Rating:** 6
**Confidence:** 5

**Summary:**

This paper studies the RLHF problem from two-player game perspective, namely the Nash learning problem with human preference, which is a generalization of the contemporary Bradley-Terry model. Unlike previous papers which learns and approximates the underlying preference model, this paper assumes only the access to query human feedback from the preference oracle. The proposed algorithm, INPO, follows a self-play training framework where each player performs a mirror descend to optimize its objective function, which consists of the win probability and a KL regularizer, preventing the current policy to deviate from reference policy. Through a carefully designed loss, the algorithm does not need to estimate the win probability of two responses. Theoretical convergence rate and intensive experiments with LLMs are provided.

**Strengths:**

1. This paper studies the Nash policy learning from the human feedback problem with only access to the query oracle, without a function approximator for the underlying preference dynamic. This setting is novel and more challenging than previous papers on similar settings. In the reviewer's view, this problem formulation also has much potential in training LLMs under diverse human populations (instead of obtaining feedback from GPT-4), where the underlying preference dynamic may be impossible to approximate.

2. The proposed algorithm INPO uses a carefully designed loss, which does not estimate the win rate of two trajectories, and has the potential to improve human query complexity. It is practical and implementable.

3. Extensive experimentation on LLMs and representative benchmarks has been conducted to show the performance of the proposed approach. INPO is shown in Table. 1 to outperform baselines of the same size in the average sense. The theoretical convergence rate is also provided, including duality gap convergence and policy convergence.

**Weaknesses:**

1. The confidence intervals in experimental results are not provided, and the significance of improvement for the proposed algorithm is not evaluated.

2. One of the claims the paper made is the algorithm does not need to estimate the average win rate of two responses, unlike previous works, say [1][2], and is potentially more query efficient. This claim is not entirely solid, as previous works, say [2], seem also implementable without estimating the average win rate accurately, i.e., set K=1 in Algorithm 1 of [2]. Even though the experiment results favor the claim compared to SPPO in Table. 1, where SPPO used more queries, more justification on where the query efficient advantage comes from needs to be detailed, i.e. is it from variance reduction, or from better-extracting information from preference?

2. The theoretical contribution is limited. The main theorems, i.e., Theorem 3 and Theorem 4, are based on infinite human queries at each iteration, and the assumption that the optimization problem (7) at each iteration can be exactly solved.
This makes the theoretical model almost the same as classic online mirror descent under convex objective function, and the proofs also directly plug in the classic OMD results. Assumption A is also unrealistic, as it requires the support of reference policy to cover all policies in the optimization trajectory, including the optimal policy. Due to these idealized assumptions, and the fact that the training only lasts for T=3 iterations in the experiment, the main theorems could not provide any insight on the empirical improvement of INPO. However, if the authors position this paper's contribution majorly on the empirical side, the level of theoretical limitation may still be acceptable.

[1] Rosset, Corby, et al. "Direct nash optimization: Teaching language models to self-improve with general preferences." arXiv preprint arXiv:2404.03715 (2024).

[2] Wu, Yue, et al. "Self-play preference optimization for language model alignment." arXiv preprint arXiv:2405.00675 (2024).

**Questions:**

1. Can you provide confidence intervals for some of your comparison results in Table 1, Table 2, and Table 3? to show the improvement of your proposed algorithm is significant compared to the benchmarks?

2. Can you provide more evidence and explanation why with the same number of human queries, your proposed algorithm can outperform DNO [1] or SPPO [2]?

3. It seems the online IPO can be viewed as a special case of Nash-MD with \beta = 0 using self-play and online mirror descent, see Sec.5 of [3]. Can you provide more comment on the difference between your algorithm, and online IPO tailored to your problem setting?

4. Can you directly optimize over equation 4 without estimating the average win rate of (y,y') accurately, i.e., use only O(1) number of y' for each y? It seems to me that this is implementable. Can you comment in more detail on why the loss transformation is necessary?

[1] Rosset, Corby, et al. "Direct nash optimization: Teaching language models to self-improve with general preferences." arXiv preprint arXiv:2404.03715 (2024).

[2] Wu, Yue, et al. "Self-play preference optimization for language model alignment." arXiv preprint arXiv:2405.00675 (2024).

[3] Calandriello, Daniele, et al. "Human alignment of large language models through online preference optimisation." arXiv preprint arXiv:2403.08635 (2024).

---

### Meta-Review · Area_Chair_2uxR · 2024-12-17

**Metareview:**

The reviewers unanimously appreciate the technical contributions of the work, although the convergence analysis is conducted for a pristine/simplified situation that is semi-disconnected from the experiments. Nonetheless, the technical developments and experimental contributions are significant, and justify acceptance.

**Additional Comments On Reviewer Discussion:**

see above

---

### Decision · Program_Chairs · 2025-01-22

Accept (Oral)